# Long term miscarriage-related hypertension and diabetes mellitus. Evidence from a United Kingdom population-based cohort study

**Kelvin Okoth**[1], **Anuradhaa Subramanian**[1], **Joht Singh Chandan**[1], **Nicola J. Adderley**[1], **G. Neil Thomas**[1], **Krishnarajah Nirantharakumar**[1☯*], **Christina Antza**[2☯]

**1** Institute of Applied Health Research, University of Birmingham, Birmingham, United Kingdom, **2** Institute of Metabolism and Systems Research, University of Birmingham, Birmingham, United Kingdom

☯ These authors contributed equally to this work.

* k.nirantharan@bham.ac.uk

## Abstract

### Background

Miscarriages affect up to a fifth of all pregnancies and are associated with substantial psychological morbidity. However, their relationship with cardiometabolic risk factors is not well known. Therefore, in this study we aimed to estimate the burden of cardiovascular risk factors including diabetes mellitus (type 1 or 2) and hypertension in women with miscarriage compared to women without a record of miscarriage.

### Methods

A population-based retrospective cohort study was conducted using IVQIA Medical Research Data UK (IMRD-UK) between January 1995 and May 2016, an anonymised electronic health records database that is representative of the UK population. A total of 86,509, 16–50-year-old women with a record of miscarriage (exposed group) were matched by age, smoking status, and body mass index to 329,865 women without a record of miscarriage (unexposed group). Patients with pre-existing hypertension and diabetes were excluded. Adjusted incidence rate ratios (aIRR) and 95% confidence intervals (95% CI) for diabetes and hypertension were estimated using multivariable Poisson regression models offsetting for person-years follow-up.

### Results

The mean age at cohort entry was 31 years and median follow up was 4.6 (IQR 1.7–9.4) years. During the study period, a total of 792 (IR 1.44 per 1000 years) and 2525 (IR 1.26 per 1000 years) patients developed diabetes in the exposed and unexposed groups, respectively. For hypertension, 1995 (IR 3.73 per 1000 years) and 1605 (IR 3.39 per 1000 years) new diagnoses were recorded in the exposed and unexposed groups, respectively. Compared to unexposed individuals, women with a record miscarriage were more likely to develop diabetes (aIRR = 1.25, 95% CI: 1.15–1.36; p<0.001) and hypertension (aIRR = 1.07, 95% CI: 1.02–1.12; p = 0.005).

**Data Availability Statement:** Data Availability: IMRD-UK data contains electronic health records from UK primary care. In compliance with the UK

Data Protection Act and licensing agreements, the data cannot be shared via a public repository. These restrictions aim to protect patient confidentiality. Access to the data is available following ethical approval from the NHS South-East Multicentre Research Ethics Committee, subject to prior independent scientific review by IMRD-UK. IMRD-UK reviews documentation for request to ensure compliance with the SRC requirements before forwarding the submission to the SRC. Requests can be submitted to MN (mustafa. dungarwalla@iqvia.com) or BN bassam. bafadhal@iqvia.com).

**Funding:** The author(s) received no specific funding for this work.

**Competing interests:** The authors have declared that no competing interests exist.

## Conclusions

Women diagnosed with miscarriage were at increased risk of developing diabetes mellitus and hypertension. Women with history of miscarriage may benefit from periodic monitoring of their cardiometabolic health.

## Introduction

Non-communicable diseases are the leading cause of mortality worldwide accounting for almost 70% of global deaths [1]. Diabetes mellitus and hypertension are some of the most important non-communicable diseases, with diabetes responsible for 4% of non-communicable deaths while 19% of the global deaths are attributable to hypertension [1,2]. Recent statistics show that 3.5 million people live with diabetes in the United Kingdom (UK), while up to 30% of the adult population have hypertension [3]. As these numbers continue to increase, identifying further risk factors for diabetes and hypertension is of great importance for early prevention and treatment.

Conventional and well-studied risk factors common to the development of cardiovascular disease and risk factors such as hypertension and diabetes include obesity, physical inactivity, family history, ethnicity, poor dietary habits, and smoking [4–7]. In addition, emerging research has shown that exposure to some obstetric complications, among them miscarriage, may be risk factors for the development of future maternal cardio-metabolic outcomes [8]. Miscarriage is defined as the spontaneous loss of pregnancy prior to foetal viability, from conception up to 24 weeks gestation, and is the most common pregnancy complication [9]. Estimates show that up to a third of all biochemical pregnancies and up to a fifth of all clinical pregnancies end up as miscarriage [9,10].

Pre-eclampsia and other obstetric complications have been included in the American Heart Association guidelines as cardiovascular risk factors for women, but miscarriage was omitted despite the fact that some of these complications share common pathological origins with miscarriage [11]. Furthermore, research through large population studies and meta-analyses has confirmed the strong association of miscarriage with cardiovascular outcomes, providing another reason for scientists to consider miscarriage in the assessment of future cardiovascular risk [12–14]. However, the link between miscarriage and the subsequent development of cardiovascular disease are not well argued or documented. There is limited evidence describing common risk factors such as family history of cardiovascular diseases, obesity, smoking and endothelial dysfunction [15–17] while other cardiovascular risk factors, such as diabetes and especially hypertension remain under explored.

Hence, the aim of this study was to conduct a population-based retrospective matched cohort to investigate the rates of diabetes and hypertension among women in the UK with incident miscarriage compared to women with an otherwise normal pregnancy who have not experienced miscarriage.

## Methods

### Study design

We conducted a population-based, retrospective open cohort study, to estimate the risk of developing diabetes (type 1 or type 2) and hypertension in women who experienced a

miscarriage (exposed) compared to those who had not (unexposed), between 1st January 1995 and 15th May 2016.

## Data source

The study used data from general practices in the UK contributing to the IMRD-UK database. The database contains more than 3 million active patients from over 700 participating general practices spread across the UK. The individuals contributing to the database are representative of the UK population [18]. The dataset consists of recorded medical diagnoses, medical prescriptions, socio-demographic details, and information from hospital admissions. Participating practices collect patient data using an electronic health record system called Vision software. Medical diagnoses and other related patient data are recorded using a hierarchical coding system called Read codes [19]. A wide range of medical conditions using IMRD-UK database have been validated, including cardiovascular diseases and diabetic conditions [20,21]. The process for selecting Read codes for the exposure and outcomes are documented in the text in S1 Table. Data extraction and cohort selection were facilitated using the Data Extraction for Epidemiological Research (DExtER) tool [22].

**Practice eligibility criteria.** Practices were eligible for inclusion from the later of the date when the practice met the acceptable mortality reporting standard and one year after the practice implemented use of the Vision software system to allow for sufficient time for the recording of important patient data [23].

## Study population

Adult women aged 16–50 years at study entry date were eligible for inclusion in the study estimating incidence rate of diabetes and hypertension. Participant entry into the study was at the latest of their 16th birthday, one year after registration with an eligible practice, or study start date.

## Exposure

Women with a Read code record of miscarriage diagnosis (exposure) were matched with up to four women without a record of miscarriage (unexposed) randomly selected from the pool of patients with a record of a previous pregnancy carried to delivery. The exposed and unexposed participants were matched by age (within 1 year), body mass index (BMI: +/-2 kg/m$^2$) and smoking status.

## Follow up period

The date of diagnosis of miscarriage served as the index date for newly diagnosed patients (incidence cases). To mitigate immortal time bias, matched unexposed patients were assigned the same index date as their corresponding exposed patients. The follow up period was defined as the time between the patient's index date and exit date. The exit date was defined as the earliest of (i) a diagnosis of type 1 or 2 diabetes or hypertension, (ii) death of patient, (iii) study end date, (iv) patient left the practice, (v) last data collection from the practice.

## Outcomes

The outcome was the incident diagnosis of either diabetes mellitus (type 1 or 2) or hypertension. The outcomes were identified using the relevant Read codes. Both outcomes are part of the quality outcomes framework (QOF) and are well recorded in primary care settings [18].

### Ethical approval

IQVIA Medical Research Data (IMRD) incorporates data from The Health Improvement Network (THIN) a Cegedim database [24]. Reference made to THIN is intended to be descriptive of the data asset licensed by IQVIA.

Anonymised data were used throughout the study provided by the data provider to the University of Birmingham. Studies using IMRD-UK database have had initial ethical approval from the NHS South-East Multicentre Research Ethics Committee, subject to prior independent scientific review. The Scientific Review Committee (IQVIA) approved the study protocol (SRC Reference Number: 17THIN075).

### Study covariates

The following potential confounders were included in the study: age, Townsend quintiles of deprivation, smoking status, BMI, lipid medication (current users), connective tissue disorders, and reproductive comorbidities (pre-eclampsia, gestational diabetes mellitus) [25].

For each of the study covariates the most recently documented record prior to study entry was used. Self-reported smoking status was categorised as never smoker, ex-smoker, current smoker or missing. BMI measured in $kg/m^2$ was categorised as underweight or normal ($<25kg/m^2$), overweight ($25–30 \ kg/m^2$), obese ($>30 \ kg/m^2$), and missing for those with missing or implausible values. A record for prescription of lipid-lowering medication was used as a proxy for high cholesterol levels. Patients prescribed lipid medication within the last sixty days prior to cohort entry were defined as current users. Connective tissue disorders (CTD) are a group of autoimmune diseases which include systemic lupus erythematosus, polymyalgia rheumatica, polymyositis, and rheumatoid arthritis.

### Statistical analysis

Continuous variables were presented as mean ± standard deviation (SD) or median and interquartile range (IQR) depending on their distribution, and proportions were used for categorical variables.

Incidence rates (IR) were calculated for each outcome separately. These were estimated as the number of incident cases of the outcome divided by the total number of person-years follow up. The person-years follow up was from the entry date to the exit date. Incidence rates were expressed as per 1000 person-years at risk.

Poisson regression models offsetting for person-years follow-up, were used to estimate incidence rate ratios (IRR) and 95% confidence intervals by making comparisons between those with a record of miscarriage (exposed) to those without a record of miscarriage (unexposed). Crude and adjusted IRRs were calculated. Patients with a record of the outcome at baseline were excluded from the corresponding analysis. For example, for the analysis examining the association between miscarriage and diabetes, patients with diabetes at baseline were excluded from the analysis. Statistical significance was set at $p<0.05$. Regression models were adjusted for age, Townsend quintile of deprivation, smoking status, BMI, polycystic ovary syndrome (PCOS), connective tissue disorders, pre-eclampsia, and gestational diabetes mellitus. Statistical analysis was performed using Stata version 14.2 (Stata Corps, College Station, Texas, USA).

## Results

### Population characteristics

Following application of inclusion/exclusion criteria, the final study population comprised 416,374 women aged between 16–50 years. Fig 1 presents the study participants flow chart.

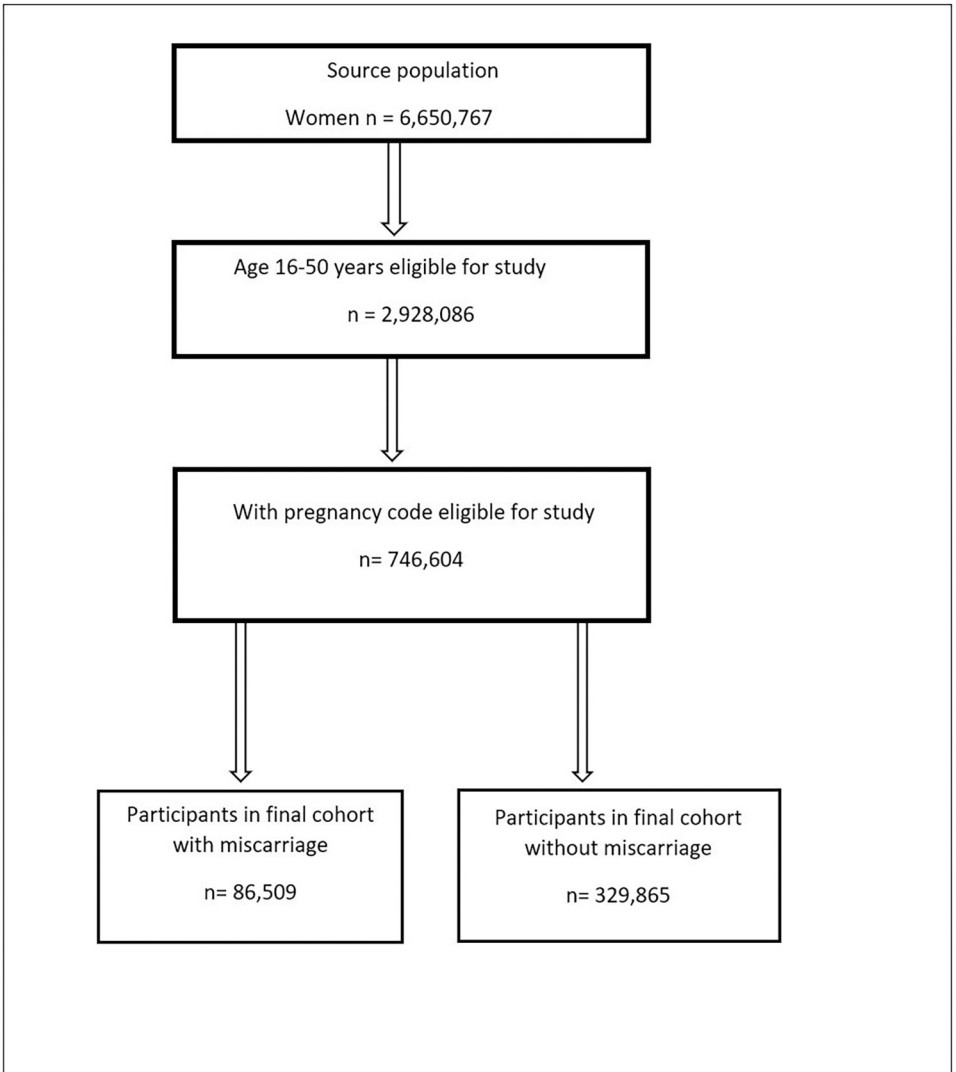

**Fig 1. Study participant flow chart.**

86,509 women with a record of miscarriage were included in the exposed population, and 329,865 women without a record of miscarriage were included in the unexposed population. The median duration of follow-up in the miscarriage cohort was 4.8 (IQR 1.9–9.4) years, while in the unexposed cohort it was 4.4 (IQR 1.6–9.1) years.

   Table 1 summarises the baseline characteristics by exposure status. Overall, the participant characteristics between the two groups were broadly similar. However, compared to women without a record of miscarriage, women with a record of miscarriage were more likely to be ex- smokers (14.2% versus 13.6%) and be diagnosed with PCOS (3.4% versus 2.9%), while they were less likely to be diagnosed with gestational diabetes (0.8% versus 1.6%).

## Diabetes mellitus

During follow up, there were a total of 792 incident diagnoses of type 1 or 2 diabetes in the miscarriage group, while 2,525 incident diagnoses of diabetes were reported in the unexposed group. The incidence rate per 1000 person-years at risk in the miscarriage cohort and the

**Table 1. Baseline characteristics of the study population.**

| | Exposed to miscarriage n = 86,509 | Unexposed to miscarriage n = 329,865 |
|---|---|---|
| | | |
| **Maternal Age, mean (SD)** | 30.9 (6.8) | 30.4 (6.5) |
| **Body Mass Index, n (%)** | | |
| <25kg/m$^2$ | 41,494 (48) | 161,716 (49) |
| **25–30 kg/m$^2$** | 16,826 (19.5) | 68,045 (20.7) |
| **>30 kg/m$^2$** | 11,398 (13.2) | 44,316 (13.4) |
| **Missing** | 16,791 (19.4) | 55,788 (16.9) |
| **Smoking status, n (%)** | | |
| **Non-smokers** | 47,146 (54.5) | 179,301 (54.4) |
| **Ex-smokers** | 12,305 (14.2) | 44,757 (13.6) |
| **Smokers (current)** | 20,948 (24.2) | 81,461 (24.7) |
| **Missing** | 6,110 (7.1) | 24,346 (7.4) |
| **Use of lipid lowering medication, n (%)** | | |
| **On medication** | 370 (0.4) | 1174 (0.4) |
| **Townsend deprivation Index, n (%)** | | |
| **1 (Least deprived)** | 17,862 (20.7) | 68,708 (20.8) |
| **2** | 15,997 (18.5) | 61,036 (18.5) |
| **3** | 17,805 (20.6) | 67,313 (20.4) |
| **4** | 16,464 (19.0) | 63,305 (19.2) |
| **5 (Most deprived)** | 12,638 (14.6) | 46,625 (14.1) |
| **Missing** | 5,743 (6.6) | 22,878 (6.9) |
| **Co-morbidities at baseline, n (%)** | | |
| **Hypertension** | 1,150 (1.3) | 4,103 (1.2) |
| **Diabetes** | 734 (0.9) | 2,181 (0.7) |
| **Ischaemic heart disease** | 31 (0.0) | 180 (0.1) |
| **Atrial fibrillation** | 21 (0.0) | 100 (0.0) |
| **Heart failure** | 13 (0.0) | 64 (0.0) |
| **Stroke/ TIA** | 108 (0.1) | 425 (0.1) |
| **Gestational diabetes** | 651 (0.8) | 5170 (1.6) |
| **Pre-eclampsia** | 400 (0.5) | 2,539 (0.8) |
| **Connective tissue disorders** | 403 (0.5) | 1,365 (0.4) |
| **Polycystic ovary syndrome** | 2,918 (3.4) | 9,581 (2.9) |

SD = standard deviation; TIA = transient ischaemic attack.

unexposed cohort was 1.44 and 1.26, respectively. Women exposed to miscarriage had a 15% higher incidence of diabetes compared to women unexposed to miscarriage (crude IRR = 1.15, 95% CI: 1.06–1.24, p = 0.001). After adjustment for age, BMI, Townsend deprivation index, smoking, lipid profile, connective tissue disorders, gestational diabetes mellitus, PCOS and pre-eclampsia, a record of miscarriage was associated with a 25% (IRR = 1.25, 95% CI: 1.15–1.36, p<0.001) higher incidence of diabetes compared to women in the unexposed group. The results are summarised in **Table 2**.

## Hypertension

During the follow up period, there were 1,995 incident diagnoses of hypertension among women with a record of miscarriage and 6,637 incident diagnoses among women without a record of miscarriage. The incidence rate per 1000 person-years at risk was 3.73 and 3.39 among women with and without a record of miscarriage, respectively. Compared to

**Table 2. Crude and adjusted incidence rate ratio for diabetes mellitus and hypertension.**

|  | Diabetes mellitus | | Hypertension | |
|---|---|---|---|---|
|  | **Exposed** | **Unexposed** | **Exposed** | **Unexposed** |
| Population | 85,775 | 327,684 | 85,359 | 325,762 |
| Events, n (%) | 792 | 2,525 | 1995 | 6637 |
| Person-years at risk | 546,599 | 2,001,260 | 535,541 | 1,959,223 |
| Incidence rate/1000 person years | 1.45 | 1.26 | 3.73 | 3.39 |
| Crude IRR (95% CI) | 1.15 (1.06–1.24) | | 1.10 (1.05–1.16) | |
| P-value | 0.001 | | <0.001 | |
| Adjusted IRR 95% CI | 1.25$^{*}$ (1.15–1.36) | | 1.07# (1.02–1.12) | |
| P-value | <0.001 | | 0.005 | |

IR = Incidence Rate, IRR = Incidence Rate Ratio.

$^{*}$ = Adjusted for: Age, BMI, Townsend deprivation index, smoking, lipid profile, connective tissue disorders, gestational diabetes mellitus, polycystic ovary syndrome, pre-eclampsia and hypertension.

# = Adjusted for: Age, BMI, Townsend deprivation index, smoking, lipid profile, connective tissue disorders, gestational diabetes mellitus, polycystic ovary syndrome, pre-eclampsia and diabetes (type 1 and 2).

unexposed women, women with a record of miscarriage were more likely to develop hypertension (crude IRR = 1.10, 95% CI: 1.05–1.16, p<0.001). This result remained statistically significant (IRR = 1.07, 95% CI: 1.02–1.12, p = 0.005) in the fully adjusted model. The results are summarised in **Table 2**.

## Discussion

The aim of this cohort study was to estimate the burden of the cardiovascular risk factors, diabetes (type 1 or 2) and hypertension, subsequent to miscarriage among women in the UK aged 16–50 years. Our analysis revealed that women with a record of miscarriage developed diabetes and hypertension at significantly higher rates compared to women without a record of miscarriage. These associations remained significant after adjustment for confounders, with exposure to miscarriage being associated with 25% and 7% higher rates of diabetes and hypertension, respectively.

There is limited published evidence describing the long-term outcomes of miscarriage relating to diabetes and hypertension. A range of pregnancy characteristics have been recorded as determinants for higher blood pressure later in a woman's life [26]. In contrast to our findings, a Finnish cross-sectional study found no association between miscarriage and the risk of developing hypertension (OR 0.8; 95% CI, 0.7–1.1) [27]. A Danish cohort study that investigated the association between pregnancy loss and subsequent risk of atherosclerotic disease among women aged 12 years and above, reported that women exposed to miscarriage had a higher risk of developing secondary, reno-vascular hypertension (adjusted IRR 1.20; 95% CI, 1.05–1.38) [28]. However, information on key confounders such as BMI and socioeconomic status were missing from the Danish study, and biased effect estimates can therefore not be ruled out. We found a significant the association between miscarriage and hypertension after adjustments for BMI, socioeconomic status, PCOS, gestational diabetes, and connective tissue disorders, conditions which may predispose to both miscarriage and hypertension. This may explain the lower effect estimates for hypertension that were reported in our study. The findings from our study are consistent the results of a US prospective cohort study which reported that women with early miscarriage (<12 weeks) had a slightly higher risk of hypertension (adjusted HR 1.06; 95% CI: 1.02–1.10) and type 2 diabetes (adjusted HR 1.20; 95% CI: 1.07–1.34) [29].

A Scottish retrospective cohort study published in 2003, investigated the association between reproductive factors, BMI and risk of diabetes. The study reported 70% (adjusted OR 1.70: 95% CI 0.82–3.52) higher odds of developing diabetes among women with a record miscarriage compared to women without a record of miscarriage after adjustment for age and BMI only [30]. More comparable to the findings of our study, a German cohort study found that women with miscarriage had a 30% (HR 1.30; 95% CI 1.01–1.68) higher risk of developing diabetes after adjustment for a number of confounders [31]. In addition to age, BMI, socioeconomic status, and hyperlipidaemia, our model was further adjusted for gestational diabetes, polycystic ovary syndrome, and connective tissue disorders, as these conditions predispose to spontaneous pregnancy loss and have been identified as risk factors for the development of diabetes [32–36]. A study in a Chinese population also found a statistically significant increase in risk of diabetes in women with miscarriage compared to women without, the effect estimate was smaller (HR 1.03; 95% CI, 1.00–1.05) compared to women without a record of miscarriage [37].

The mechanisms linking miscarriage to future maternal risk of cardiometabolic outcomes are unclear but may involve endothelial dysfunction, genetics, and metabolic syndrome. Adverse pregnancy outcomes (APOs) including pre-eclampsia, gestational diabetes, miscarriage, and pre-term birth are linked to future maternal risk of cardiometabolic complications [25,35,38]. The common underlying etiology of these APOs are placental anomalies triggered by endothelial dysfunction. It is postulated that endothelial dysfunction persists for several years beyond APOs leading to future risk of vascular complications [16]. There is an association between miscarriage and family history of cardiovascular complications suggesting that there are shared genetic anomalies. A study by Smith et al noted that women with history of two or more pregnancy losses prior to their first delivery had higher risk of family history of coronary heart disease [17]. Women with history of miscarriage may have an underlying susceptibility to metabolic syndrome, a condition that manifests with a combination of high blood glucose, hypertension, obesity and dyslipidemia [39].

This study provides evidence describing the incidence of diabetes and hypertension after miscarriage compared to women with normal pregnancies in a large, well-powered cohort study that adjusted for a wide range of well-defined risk factors. However, despite the strengths of the study, the results should be interpreted taking into account its limitations. The study explored diabetes as a broad endpoint, therefore specific association between miscarriage and either type 1 or type 2 diabetes was not evaluated. Misclassification of the exposure (miscarriage) or the outcome (diabetes or hypertension) may occur as some patients may be undiagnosed or asymptomatic. We do not expect there to be a systematic difference in the recording of the outcomes among women with and without miscarriage. However, the effect of including the exposure or the outcome in the wrong cohort is not clear and may not necessarily bias the effect estimate towards the null [40]. Furthermore, miscarriage maybe under recorded in primary care and therefore the number of incident cases may be underestimated. Although our study shows an association between history of miscarriage and future maternal risk of diabetes or hypertension, there is potential for unmeasured confounding for example from factors such as pregnancy weight gain or micronutrient deficiencies which are poorly recorded in UK primary care.

In terms of clinical practice, results from this study suggest that women with a history of miscarriage might be considered for referral for hypertension and diabetes screening and follow up. Current diabetes and hypertension guidelines do not mention this population as a high-risk population for monitoring, but our evidence in combination with the previous literature suggesting that this population is also at higher risk for cardiovascular outcomes indicate

that these risk factors should be considered by clinicians [41,42]. Future research should be focused on the mechanism behind miscarriage and cardiometabolic risk.

## Conclusion

To conclude, higher rates of diabetes and hypertension were observed among women exposed to miscarriage compared to those unexposed to miscarriage. Given that diabetes and hypertension are leading causes of morbidity and mortality, women with history of miscarriage may benefit from periodic monitoring of their cardiometabolic health. Future studies are required to confirm findings from the present study, clarify the biological mechanism behind the observed associations, and investigate whether interventions that prevent miscarriage will be effective in reducing the excess burden of diabetes and hypertension among women with miscarriage.

## Supporting information

**S1 Table. Search strategy based to read codes.**
(DOCX)

## Acknowledgments

Prior publication of the study in abstract form: No Transparency declaration: The lead author (the manuscript's guarantor) affirms that this manuscript is an honest, accurate, and transparent account of the study being reported; that no important aspects of the study have been omitted; and that any discrepancies from the study as planned (and, if relevant, registered) have been explained.

## Author Contributions

**Conceptualization:** Kelvin Okoth, Joht Singh Chandan, Krishnarajah Nirantharakumar.

**Formal analysis:** Kelvin Okoth, Anuradhaa Subramanian, Joht Singh Chandan, Christina Antza.

**Investigation:** Kelvin Okoth.

**Methodology:** Kelvin Okoth.

**Software:** Anuradhaa Subramanian, Joht Singh Chandan, Nicola J. Adderley, G. Neil Thomas.

**Supervision:** Joht Singh Chandan, Nicola J. Adderley, G. Neil Thomas, Krishnarajah Nirantharakumar, Christina Antza.

**Writing – original draft:** Kelvin Okoth, Christina Antza.

**Writing – review & editing:** Anuradhaa Subramanian, Joht Singh Chandan, Nicola J. Adderley, G. Neil Thomas, Krishnarajah Nirantharakumar, Christina Antza.

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
