## [Decision Letter · Decision Letter 0]

31 Aug 2021

PONE-D-21-18971

Long term miscarriage-related hypertension and diabetes mellitus. Evidence from a United Kingdom population-based cohort study.

PLOS ONE

Dear Dr. Okoth,

Thank you for submitting your manuscript to PLOS ONE. After careful consideration, we feel that it has merit but does not fully meet PLOS ONE’s publication criteria as it currently stands. Therefore, we invite you to submit a revised version of the manuscript that addresses the points raised during the review process.

The manuscript and the reviewers’ comments were carefully evaluated. The manuscript was appreciated by the Reviewers. Nevertheless, as suggested, the manuscript requires extensive improvement before to be considered for publication. Further suggested revisions are in detail reported in the Reviewers’ comments.

We look forward to receiving your revised manuscript.

Kind regards,

Simone Garzon

Academic Editor

PLOS ONE

Journal Requirements:

Reviewers' comments:

Reviewer's Responses to Questions

**Comments to the Author**

1. Is the manuscript technically sound, and do the data support the conclusions?

Reviewer #1: Yes

Reviewer #2: Yes

2. Has the statistical analysis been performed appropriately and rigorously? 

Reviewer #1: Yes

Reviewer #2: Yes

3. Have the authors made all data underlying the findings in their manuscript fully available?

Reviewer #1: No

Reviewer #2: Yes

4. Is the manuscript presented in an intelligible fashion and written in standard English?

Reviewer #1: Yes

Reviewer #2: Yes

5. Review Comments to the Author

Reviewer #1: Dear editor,

Thank you for your invitation of reviewing the manuscript entitled ”Long term miscarriage-related hypertension and diabetes mellitus. Evidence from a United Kingdom population-based cohort study”. This study investigated the burden of the cardiovascular risk factors hypertension and diabetes mellitus (type 1 or 2) in women with and without a record of miscarriage. It found that women diagnosed with miscarriage were at increased risk of developing diabetes mellitus and hypertension.

The analysis method of this paper is correct and the text is fluent. But there are several very important issues that need to be further considered, such as how to consider the condition of having hypertension and diabetes before pregnancy. This has a great influence on the results and conclusions of this study. I hope the authors can respond to the following questions first, and then consider further. My comments to the article are listed below:

1.Page 2, line 37-39, the sentence “Therefore, in this study we aimed to the burden of the cardiovascular risk factors hypertension and diabetes mellitus (type 1 or 2) in women with and without a record of miscarriage” needs revision.

2.Table 1, “Smoking status, n (%)” should be bold.

3.Please pay attention to the formatting of references, such as letter case and the full name and abbreviation of the journal (eg Reference 38)

4.Are there people with diabetes who had hypertension before pregnancy in this study? Or in other words, how to define the relationship between the occurrence of hypertension and diabetes in this study and the miscarriage of this pregnancy?

5.Was weight gain during pregnancy considered in this study, because weight gain during pregnancy is also closely related to the occurrence of diabetes and hypertension.

6.What do you think is the possible mechanism for miscarriage to increase the risk of high blood pressure and diabetes? Please describe in the discussion.

7.Should the conclusion of this study include the importance of preventing preterm birth? What are the main conclusions and significance of this research? And modify it in the text.

8.When referring to hypertension and diabetes in this study, sometimes it is “hypertension and diabetes”, sometimes it is “diabetes and hypertension”, please be consistent.

Reviewer #2: The research inference comes from the medical record, but it does not deal with the possible bias from the record. What percentage of subjects are pregnant or have miscarriages that are not recorded in the medical records? Is there anyone who has developed signs of diabetes or hypertension, but has not yet started medical treatment and therefore has not entered a diagnosis? If there are any of the above, the impact on the results should be taken into consideration.

6. PLOS authors have the option to publish the peer review history of their article (what does this mean?). If published, this will include your full peer review and any attached files.

Reviewer #1: No

Reviewer #2: No

---

## [Author Response · Author response to Decision Letter 0]

30 Oct 2021

14th October 2021

Dear Dr Garzon,

Thank you for considering our manuscript, ‘Long term miscarriage-related diabetes mellitus and hypertension. Evidence from a United Kingdom population-based cohort study.’, for publication in PLOS ONE.

We thank the reviewers for their comments and suggestions, and we respond as follows.

Reviewer 1

1.Page 2, line 37-39, the sentence “Therefore, in this study we aimed to the burden of the cardiovascular risk factors hypertension and diabetes mellitus (type 1 or 2) in women with and without a record of miscarriage” needs revision.

Thank you. We have revised line 37-39 to read as follows “Therefore, in this study we aimed to estimate the burden of cardiovascular risk factors including diabetes mellitus (type 1 or 2) and hypertension among women with miscarriage compared to those without miscarriage.”

2.Table 1, “Smoking status, n (%)” should be bold.

The text “Smoking status, n (%)” in table 2 is in bold in the revised manuscript.

3.Please pay attention to the formatting of references, such as letter case and the full name and abbreviation of the journal (eg Reference 38)

We have revised reference 38 (Reference 41 in revised manuscript) as below

Williams B, Mancia G, Spiering W, Rosei EA, Azizi M, Burnier M, et al. 2018 ESC Scientific Document Group, ESC/ESH Guidelines for the management of arterial hypertension: The Task Force for the management of arterial hypertension of the European Society of Cardiology (ESC) and the European Society of Hypertension (ESH) Vol. 39, European Heart Journal. Oxford University Press; 2018. p. 3021–104. 

We have also checked that all the other references are cited appropriately.

4.Are there people with diabetes who had hypertension before pregnancy in this study? Or in other words, how to define the relationship between the occurrence of hypertension and diabetes in this study and the miscarriage of this pregnancy?

Yes, there were participants who had various comorbidities including diabetes and hypertension before pregnancy. We have presented this information in the baseline table (Table 1). We excluded patients with a record of the outcome (diabetes or hypertension) at baseline, from the corresponding univariable and multivariable analysis. For instance, for the analysis examining the association between miscarriage and diabetes, patients with diabetes at baseline were excluded from the analysis. On the other hand, in the analysis examining the association between miscarriage and hypertension patients with hypertension at baseline were excluded from the analysis. We have now clarified this in the manuscript; please see line 192 to 194 of the revised manuscript. 

In the revised manuscript, we have clarified the covariates that were included in the multivariable model as follows 

The model (*) examining the relationship between miscarriage and diabetes adjusted for age, BMI, Townsend deprivation index, smoking, lipid profile, connective tissue disorders, gestational diabetes mellitus, polycystic ovary syndrome, pre-eclampsia, and hypertension. The model (#) examining the relationship between miscarriage and hypertension adjusted for age, BMI, Townsend deprivation index, smoking, lipid profile, connective tissue disorders, gestational diabetes mellitus, polycystic ovary syndrome, pre-eclampsia, and diabetes (type 1 and 2).

5.Was weight gain during pregnancy considered in this study, because weight gain during pregnancy is also closely related to the occurrence of diabetes and hypertension. 

We agree with the reviewer that weight gain during pregnancy is associated with an increased risk of diabetes and hypertension. Information on pregnancy weight gain is poorly recorded in UK primary care since there is poor linkage of health records among the various services (local maternity units, GP surgery, children centers, and hospitals) that provide ante-natal care.

We have included the following statement in the limitation section to address this issue

“Although our study shows an association between history of miscarriage and future maternal risk of diabetes or hypertension, there is potential for unmeasured confounding for example from factors such as pregnancy weight gain or micronutrient deficiencies which are poorly recorded in UK primary care.” 

6.What do you think is the possible mechanism for miscarriage to increase the risk of high blood pressure and diabetes? Please describe in the discussion.

The mechanistic links behind the association between miscarriage and future risk of diabetes and hypertension are not clear but may involve endothelial dysfunction, genetics or metabolic syndrome.

We have included the following paragraph in the revised manuscript as suggested.

“Biological plausibility”

“The mechanisms linking miscarriage to future maternal risk of cardiometabolic outcomes are unclear but may involve endothelial dysfunction, genetics and metabolic syndrome. Adverse pregnancy outcomes (APOs) including pre-eclampsia, gestational diabetes, miscarriage, and pre-term birth are linked to future maternal risk of cardiometabolic complications.(1–3) The common underlying aetiology of these APOs are placental anomalies triggered by endothelial dysfunction. It is postulated that endothelial dysfunction persists for several years beyond APOs leading to future risk of vascular complications.(4) Endothelial dysfunction is also associated later risk of type 2 diabetes mellitus after adjustment for known risk factors.(5) Therefore, endothelial dysfunction may potentially precede future development of both diabetes and hypertension among women with history of miscarriage. There is an association between miscarriage and family history of cardiovascular complications suggesting that there are shared genetic anomalies. A study by Smith et al noted that women with history of two or more pregnancy losses prior to their first delivery had higher risk of family history of coronary heart disease.(6) Women with history of miscarriage may have an underlying susceptibility to metabolic syndrome, a condition that manifests with a combination of high blood glucose, hypertension, obesity and dyslipidemia.(7)”

7.Should the conclusion of this study include the importance of preventing preterm birth? What are the main conclusions and significance of this research? And modify it in the text.

We agree that the conclusion of our study should highlight the importance of preventing miscarriage. The mechanistic links between miscarriage and poor cardiometabolic health are unclear. Miscarriage may be the initial event that triggers poor cardiometabolic health or it may serve as a stress test that unmasks women already predisposed to cardiometabolic complications. Whether preventing miscarriage will translate to a reduced risk of diabetes and hypertension is not clear and should be the subject of future research. 

We have now expanded the conclusion to read as follows

“To conclude, higher rates of diabetes and hypertension were observed among women exposed to miscarriage compared to those unexposed to miscarriage. Given that diabetes and hypertension are leading causes of morbidity and mortality, women with history of miscarriage may benefit from periodic monitoring of their cardiometabolic health. Future studies are required to confirm findings from the present study, clarify the biological mechanism behind the observed associations, and investigate whether interventions that prevent miscarriage will be effective in reducing the excess burden of diabetes and hypertension among women with miscarriage.”

8.When referring to hypertension and diabetes in this study, sometimes it is “hypertension and diabetes”, sometimes it is “diabetes and hypertension”, please be consistent.

We thank the reviewer for bringing this to our attention. Diabetes and hypertension are now referred to consistently in the revised manuscript.

Reviewer 2

The research inference comes from the medical record, but it does not deal with the possible bias from the record. What percentage of subjects are pregnant or have miscarriages that are not recorded in the medical records? Is there anyone who has developed signs of diabetes or hypertension, but has not yet started medical treatment and therefore has not entered a diagnosis? If there are any of the above, the impact on the results should be taken into consideration.

We thank the reviewer for their comment. The proportion of subjects that were pregnant or had miscarriage that was recorded in UK primary care is not entirely clear. A survey of maternity services in the UK revealed that 96% of pregnant women had contacted a healthcare professional (midwife or general practitioner) by the time they were 12 weeks pregnant. The majority of women (66%) contacted their general practitioner. For women who are pregnant for the first time the median number of antenatal care visits was 9 while for women with a previous pregnancy it was 8.(8) We believe a substantial proportion of miscarriage cases will be documented in primary care. With regard to undiagnosed outcomes (diabetes and hypertension), 5.5 million people in England live with undiagnosed hypertension while 1 million people in the UK live with undiagnosed diabetes.(9,10)

We agree with the reviewer that there is potential for misclassification bias as some women with the exposure (miscarriage) and the outcome (diabetes and hypertension) may be missed. We have added the following paragraph under the limitations section. 

“Misclassification of the exposure (miscarriage) or the outcome (diabetes or hypertension) may occur as some patients may be undiagnosed or asymptomatic. We do not expect there to be a systematic difference in the recording of the outcomes among women with and without miscarriage. However, the effect of including the exposure or the outcome in the wrong cohort is not clear and may not necessarily bias the effect estimate towards the null.(11)”

If you have any further queries, please do not hesitate to contact us.

Yours sincerely,

Kelvin Okoth on behalf all authors 

References

1. Okoth K, Chandan JS, Marshall T, Thangaratinam S, Thomas GN, Nirantharakumar K, et al. Association between the reproductive health of young women and cardiovascular disease in later life: Umbrella review. Vol. 371, The BMJ. BMJ Publishing Group; 2020. 

2. Li S, Zhang M, Tian H, Liu Z, Yin X, Xi B. Preterm birth and risk of type 1 and type 2 diabetes: systematic review and meta-analysis. Obes Rev. 2014 Oct 1;15(10):804–11. 

3. Wu P, Kwok CS, Haththotuwa R, Kotronias RA, Babu A, Fryer AA, et al. Pre-eclampsia is associated with a twofold increase in diabetes: a systematic review and meta-analysis. Diabetologia. 2016 Dec 1;59(12):2518–26. 

4. Germain AM, Romanik MC, Guerra I, Solari S, Reyes MS, Johnson RJ, et al. Endothelial Dysfunction. Hypertension. 2007 Jan;49(1):90–5. 

5. JB M, FB H, N R, JE M. Biomarkers of endothelial dysfunction and risk of type 2 diabetes mellitus. JAMA. 2004 Apr 28;291(16):1978–86. 

6. Smith G, Wood A, Pell J, Hattie J. Recurrent miscarriage is associated with a family history of ischaemic heart disease: a retrospective cohort study. BJOG An Int J Obstet Gynaecol. 2011 Apr;118(5):557–63. 

7. Sharma S, Yadav S, Chandiok K, Sharma RS, Mishra V, Saraswathy KN. Protein signatures linking history of miscarriages and metabolic syndrome: a proteomic study among North Indian women. PeerJ. 2019;7(2). 

8. The National Perinatal Epidemiology Unit. Safely delivered : a national survey of women’s experience of maternity care 2014. 2015. 

9. CA W, S O, RIG H. Diabetes in the UK: 2019. Diabet Med. 2020 Feb 1;37(2):242–7. 

10. CVD prevention: detecting and treating hypertension [Internet]. [cited 2021 Oct 13]. Available from: https://stpsupport.nice.org.uk/cvd-prevention-hypertension/index.html

11. Brakenhoff TB, Van Smeden M, Visseren FLJ, Groenwold RHH. Random measurement error: Why worry? An example of cardiovascular risk factors. PLoS One. 2018;13(2):1–8.

---

## [Decision Letter · Decision Letter 1]

10 Dec 2021

Long term miscarriage-related hypertension and diabetes mellitus. Evidence from a United Kingdom population-based cohort study.

PONE-D-21-18971R1

Dear Dr. Okoth,

We’re pleased to inform you that your manuscript has been judged scientifically suitable for publication and will be formally accepted for publication once it meets all outstanding technical requirements.

Kind regards,

Simone Garzon

Academic Editor

PLOS ONE

Additional Editor Comments (optional):

Reviewers' comments:

Reviewer's Responses to Questions

**Comments to the Author**

1. If the authors have adequately addressed your comments raised in a previous round of review and you feel that this manuscript is now acceptable for publication, you may indicate that here to bypass the “Comments to the Author” section, enter your conflict of interest statement in the “Confidential to Editor” section, and submit your "Accept" recommendation.

Reviewer #2: All comments have been addressed

2. Is the manuscript technically sound, and do the data support the conclusions?

Reviewer #2: Yes

3. Has the statistical analysis been performed appropriately and rigorously? 

Reviewer #2: Yes

4. Have the authors made all data underlying the findings in their manuscript fully available?

Reviewer #2: Yes

5. Is the manuscript presented in an intelligible fashion and written in standard English?

Reviewer #2: Yes

6. Review Comments to the Author

Reviewer #2: All comments have been addressed.

According to the reply, the bias should be limited.

No further recommendation to be given.

7. PLOS authors have the option to publish the peer review history of their article (what does this mean?). If published, this will include your full peer review and any attached files.

Reviewer #2: No

---

## [Editor Report · Acceptance letter]

15 Dec 2021

PONE-D-21-18971R1 

Long term miscarriage-related hypertension and diabetes mellitus. Evidence from a United Kingdom population-based cohort study. 

Dear Dr. Okoth:

I'm pleased to inform you that your manuscript has been deemed suitable for publication in PLOS ONE. Congratulations! Your manuscript is now with our production department. 

Kind regards, 

on behalf of

Dr. Simone Garzon 

Academic Editor

PLOS ONE